# Large language models transition from integrating across position-yoked, exponential windows to structure-yoked, power-law windows

**David Skrill**
Department of Biostatistics and Computational Biology
University of Rochester Medical Center
Rochester, NY 14642
`david_skrill@urmc.rochester.edu`

**Sam V. Norman-Haignere**
Depts. of Biostatistics and Computational Biology, Neuroscience
University of Rochester Medical Center
Depts. of Brain and Cognitive Sciences, Biomedical Engineering
University of Rochester
Rochester, NY 14642
`samuel_norman-haignere@urmc.rochester.edu`

## Abstract

Modern language models excel at integrating across long temporal scales needed to encode linguistic meaning and show non-trivial similarities to biological neural systems. Prior work suggests that human brain responses to language exhibit hierarchically organized "integration windows" that substantially constrain the overall influence of an input token (e.g., a word) on the neural response. However, little prior work has attempted to use integration windows to characterize computations in large language models (LLMs). We developed a simple word-swap procedure for estimating integration windows from black-box language models that does not depend on access to gradients or knowledge of the model architecture (e.g., attention weights). Using this method, we show that trained LLMs exhibit stereotyped integration windows that are well-fit by a convex combination of an exponential and a power-law function, with a partial transition from exponential to power-law dynamics across network layers. We then introduce a metric for quantifying the extent to which these integration windows vary with structural boundaries (e.g., sentence boundaries), and using this metric, we show that integration windows become increasingly yoked to structure at later network layers. None of these findings were observed in an untrained model, which as expected integrated uniformly across its input. These results suggest that LLMs learn to integrate information in natural language using a stereotyped pattern: integrating across position-yoked, exponential windows at early layers, followed by structure-yoked, power-law windows at later layers. The methods we describe in this paper provide a general-purpose toolkit for understanding temporal integration in language models, facilitating cross-disciplinary research at the intersection of biological and artificial intelligence.

37th Conference on Neural Information Processing Systems (NeurIPS 2023).

# 1 Introduction

Natural language is hierarchically structured at many scales (e.g., phrases, sentences, and narrative structures), and the temporal extent of these structures is highly variable. Intelligent systems must therefore have mechanisms to flexibly integrate across diverse and variable temporal scales to infer meaning from linguistic inputs (Hochreiter and Schmidhuber [1996], Tallec and Ollivier [2018], Mahto et al. [2020]). Large language models (LLMs) excel at this task as demonstrated by their impressive performance on a wide range of linguistic benchmarks (Radford et al. [2019], Devlin et al. [2019], Brown et al. [2020], Raffel et al. [2020], Chowdhery et al. [2022], Touvron et al. [2023]). Recent studies have shown that features from these models are also highly predictive of human brain responses in high-level language regions of the human cortex (Schrimpf et al. [2021], Jain et al. [2023]), suggesting that LLMs may be powerful tools for understanding and modeling temporal integration in the brain (Caucheteux et al. [2021b]).

Prior work has shown that brain responses to speech and language exhibit structured "integration windows" that substantially constrain the influence of an input token (e.g., word) on the neural response (**Fig. 1A**) (Theunissen and Miller [1995], Lerner et al. [2011], Chien and Honey [2020], Norman-Haignere et al. [2022]). Integration windows have been studied for decades in the neuroscience literature and are thought to be central to the hierarchical organization of the brain (Hickok and Poeppel [2007], Sharpee et al. [2011], Lerner et al. [2011], Norman-Haignere et al. [2022]). By comparison, much less is known about whether LLMs exhibit structured integration windows, which is relevant for understanding the mechanisms used by these models to integrate across the multi-scale structure of language, and whether these mechanisms bear any resemblance to those used by the brain to encode linguistic meaning.

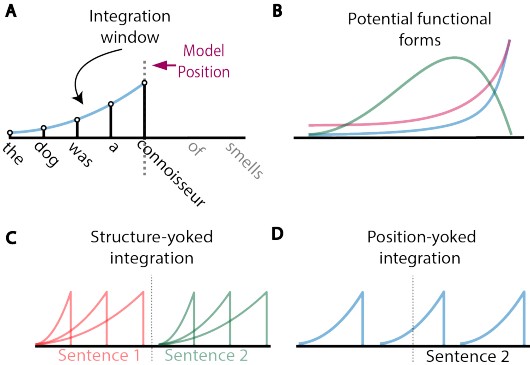

**Figure 1: Schematic of integration windows**. **A**, Illustration of a causal, exponentially distributed integration window where nearby words in the past have a greater influence on the response. **B**, Example integration windows with different functional forms: exponential (blue), power-law (pink), gamma-distributed (green), and uniform (brown). **C**, A structure-yoked integration window that dynamically contours to linguistic boundaries (e.g., sentence breaks). **D**, A position-yoked window that is unaffected by structural boundaries.

Modern attention-based architectures can in principle integrate uniformly across their entire input (Vaswani et al. [2017]). However, as we demonstrate, training can alter the effective integration window of a network and cause that window to differ substantially from the integration window implied by a model's architecture (Luo et al. [2016], Keshishian et al. [2021]), and little prior work has attempted to empirically measure the integration window of trained LLMs. To address this gap, we developed a set of general-purpose methods for estimating integration windows from black-box language models and applied these methods to characterize LLM integration windows. We focused initially on GPT-2 (Radford et al. [2019]) because it is popular, publicly available, has shown good brain predictivity (Schrimpf et al. [2021], Jain et al. [2023]), and, like the brain, is causal. We assessed the generality of our findings by repeating our analyses for two other LLMs (LLaMA and roBERTa; Liu et al. [2019], Touvron et al. [2023]) and varying important properties of our analysis.

Our paper makes the following contributions:

- We introduce a word-swap procedure for systematically measuring integration windows from language models, which does not depend on any information about the model architecture.

- We show that LLMs exhibit stereotyped integration windows that substantially constrain the influence of an input token and follow a simple function form that is well-fit by a convex combination of an exponential and a power-law function.

- Using these convex fits, we show that that integration windows expand substantially in later network layers driven in part by a shift from exponential to power-law dynamics.

- We introduce a metric for quantifying the degree to which integration windows vary with structural boundaries in language.
- Using this metric, we show that there is a substantial increase in structure-yoked integration across the layers of LLMs.
- We show that these properties (1) are present across three different LLM models (2) generalize across different structural lengths, types, and boundaries and (3) are absent from an untrained model, demonstrating that they are learned from the structure of natural language.

Code implementing our methods and corresponding analyses is available at `https://github.com/dskrill/TemporalIntegration.git`.

## 2 Relationship with prior work

A large body of prior work has focused on characterizing attention weights in LLMs, and how these weights vary with linguistic structure, word position, and network layer (Clark et al. [2019], Vig and Belinkov [2019],Abnar and Zuidema [2020]). Multiple studies have observed that attention weights align with syntactic and semantic information, with lower layers preferentially encoding local syntactic structure and higher layers encoding more complex semantic relationships (Tenney et al. [2019]). Tenney et al. [2019] demonstrated that LLMs effectively rediscover the classical NLP pipeline, with regions of the trained networks corresponding to part-of-speech (POS) tagging, parsing, named entity recognition (NER), semantic roles, and coreference. There is also evidence that the range of attended-to tokens expands from lower to higher network layers (Vig and Belinkov [2019]), potentially increasing the timescale of analysis.

However, none of these prior studies have assessed whether LLMs exhibit an integration window that constrains the influence of an input token on a unit's response. Indeed, it was not obvious *a priori* whether LLMs would have an integration window at all, given that transformer models can in principle integrate uniformly over their entire input. Moreover, attention-based metrics are limited to transformer models and thus cannot be used to evaluate and compare machine learning models with distinct architectures (e.g., LSTMs) or compare biological and artificial systems. By comparison, our methods are applicable to any language model, and using these methods, we show that LLMs exhibit integration windows that substantially constrain the influence of an input token on unit responses, vary substantially across layers, obey a simple functional form, and vary with structural boundaries at later network layers.

Prior empirical and theoretical work has characterized temporal dependencies in natural language (Chomsky [1965]. For example, the mutual information between English words declines with distance according to a power-law (Lin and Tegmark [2016]), and there have been efforts to incorporate this type of power-law structure into recurrent neural network language models (Tallec and Ollivier [2018], Mahto et al. [2020]). However, little is known about how these temporal dependencies shape integration dynamics in modern LLMs, and whether the different layers of LLMs obey power-law dynamics or some other functional form (e.g., exponential decay).

Keshishian et al. [2021] studied integration windows in automatic speech recognition (ASR) models trained to transcribe speech from audio. This study found that integration windows at late network layers of DeepSpeech2 (Amodei et al. [2015]) expanded and contracted when the audio waveform was physically stretched or compressed, suggesting its integration windows reflect structure duration. This study however only investigated time-domain audio models, and as a consequence, many of the methods and findings are not applicable to language models.

## 3 Measuring overall integration windows with word swaps

An integration window, by definition, constrains the influence of an input token depending on the position of that token (Theunissen and Miller [1995], Norman-Haignere et al. [2022]) (**Fig. 1A**). For a causal model, such as GPT-2, the integration window extends backwards from the position of the model response since only tokens that precede the model response can influence it. We sought to characterize the functional form of LLM integration windows (**Fig. 1B**) and test whether they dynamically vary with linguistic structures (**Fig. 1C**) or, alternatively, are a fixed function of relative distance between the token and model response (**Fig. 1D**).

## 3.1 Word swap procedure

We measured the response of model units to two input sequences with a single swapped word (**Fig. 2A**). For each unit, we then calculated the magnitude of the response difference between the two sequences at each model position, as well as the distance between the position of the swapped word and the position of the model response ($\Delta$). The integration window was calculated as the distant-dependent change in response magnitudes, averaged across many sequences and words swaps (**Fig. 2A**). Word swaps were sampled from a list of the 5 most probable words given the context (excluding the actual word) as computed by BigBird roBERTa Zaheer et al. [2021] (masking out the target word to be swapped).

To formalize this idea, denote the $j$th word of the $i$th sequence as $w_i[j]$ and the paired sequence with the $k$th word swapped as $w_i^{k*}[j]$. The corresponding activation sequences for a single model unit are denoted by $a_i[j]$ and $a_i^{k*}[j]$ (where $a_i[j] = f(w_i)[j]$, $a_i^{k*}[j] = f(w_i^{k*})[j]$ and $f(\cdot)$ is given by a language model that maps an input sequence to the response of a single unit; the unit index is omitted to simplify notation since all analyses are applied separately to each unit). The integration window of the unit is defined as ($I$ and $K$ are the number of sequences and swaps, respectively):

$$\theta[\Delta] = \frac{1}{IK} \sum_{i=1}^{I} \sum_{k=1}^{K} |a_i[k+\Delta] - a_i^{k*}[k+\Delta]| \tag{1}$$

Because the scale of $\theta[\Delta]$ is arbitrary, we normalized it by its maximum value which empirically always occurred at distance 0:

$$\theta_{norm}[\Delta] = \frac{\theta[\Delta]}{\theta[0]} \tag{2}$$

## 3.2 Experiments

We applied this procedure to measure integration windows from GPT-2 (using the smallest version of GPT-2 with 124M parameters, 12 layers, and 12 attention heads per layer; embedding dimension set to 768; vocabulary size equal to 50,256). We used sequences of 600 words extracted from the Brown Corpus (Francis and Kucera [1979]) (accessed using the NLTK library; Loper and Bird [2002]). In all of our experiments, we removed punctuation, capitalization and any tokens that delineated structural boundaries (e.g., [CLS], [SEP]). We also excluded sequences that contain non-alphabetical characters.

We found that the influence of a word depends strongly on the distance between the position of the model and the swapped word (**Fig. 2B**). This distance-dependent effect varied substantially across layers, with consistently longer integration windows in later network layers, suggesting that LLMs integrate hierarchically across linguistic structure, first analyzing short-term structure and then longer-term temporal structure.

To characterize the shape of these integration windows, we fit a wide variety of functional forms, motivated by the prior literature characterizing heavy-tailed distributions (Newman [2005], Mitzenmacher [2003], Altmann et al. [2009]) (equations of all forms tested are given in the appendix). Fitting was performed by minimizing the mean-squared error between the measured and predicted integration window, using the `curve_fit` function from the Scipy library (Virtanen et al. [2020]). Model accuracy was measured as the mean-squared error on test data, separate from that used to fit the model. Results were similar using other performance metrics (Kolmogorov-Smirnov (KS) test-statistic and Bayesian Information Criterion).

We found that a simple, 3-parameter functional form - a convex combination of an exponential and power-law - provided the best fit across all layers tested (**Fig. 2C**):

$$\theta_{norm}[\Delta] \approx c(\Delta+1)^{-a} + (1-c)e^{-b\Delta} \tag{3}$$

where $c$ is the convex combination parameter (constrained to lie between 0 and 1) that controls the relative contribution of the power-law and the exponential function, and the parameters $a$ and $b$ control the rate of decline for these two functions, respectively, with higher values indicating faster decline and thus shorter integration times ($a$ and $b$ were constrained to take values between 0 and 10).

For GPT-2, we found that all of the model parameters changed substantially across layers (**Fig. 2E**). The convex combination parameter ($c$) increased nearly three-fold from layer 1 (median value across

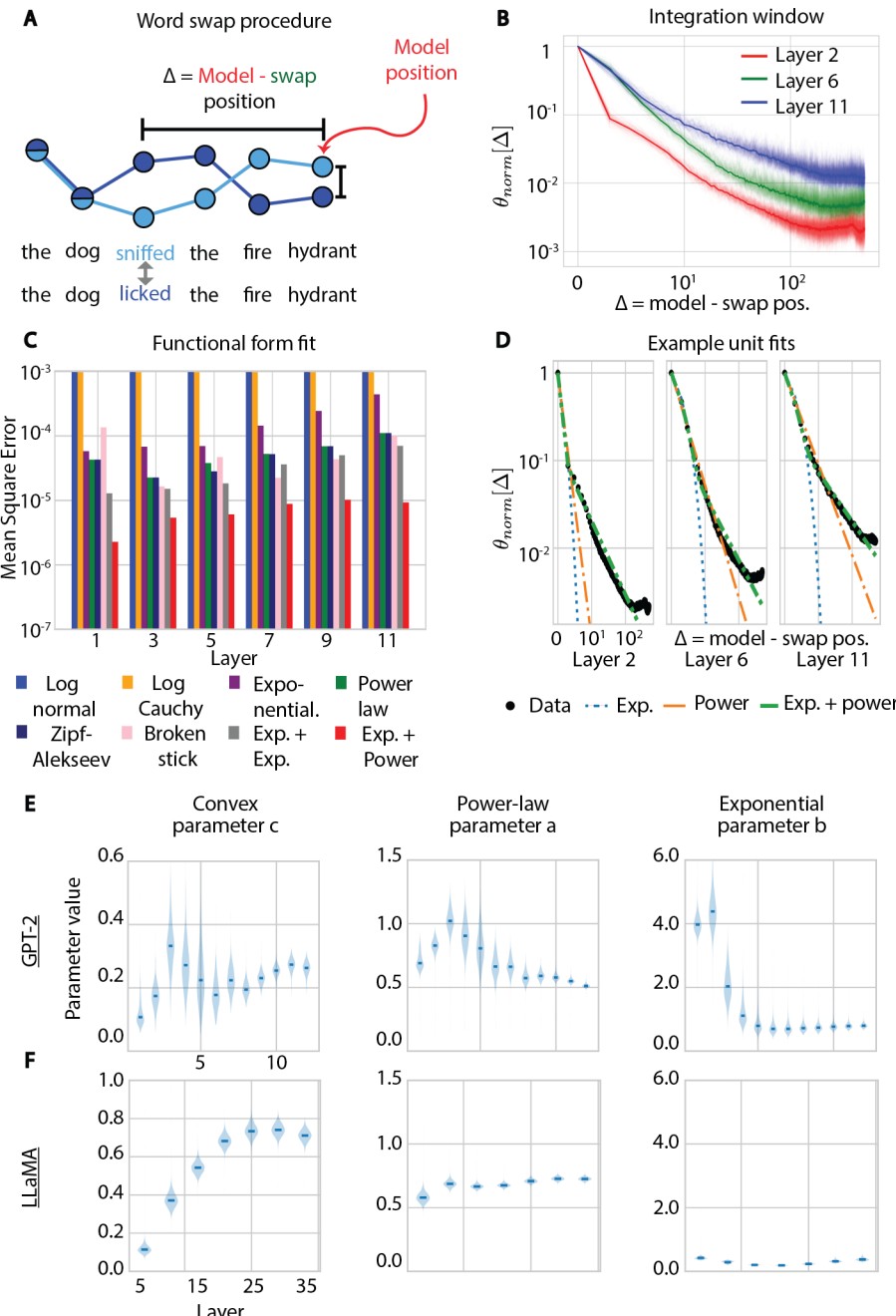

**Figure 2: Characterizing the functional form of LLM integration windows**. **A**, Schematic illustration of a unit response to two word sequences (blue and light circles) with a single word swapped. Unit responses differ after the swap due to causality. Integration windows are calculated by computing the magnitude of the response difference (vertical bracket) as a function of the distance between the position of the swapped word and the position of the model response ($\Delta$, horizontal bracket), averaged across many sequences and swaps (eq. 1). **B**, Integration windows for individual units from several layers of GPT-2 with the median across units overlaid. The influence of a word swap decreases sharply with distance (note logarithmic scale). **C**, Quantification of the goodness-of-fit for a variety of functional forms. The exponential-power fits were better across all layers. **D**, Integration windows for several example units fit using an exponential, power-law, or convex combination of both (see eq. 3). **E**, Violin plots showing the distribution of fit parameter values across all units from different layers. The convex combination parameter ($c$, left) reveals a transition from exponential to power-law dynamics across layers, and the power-law ($a$, middle) and exponential rate ($b$, right) parameters reveal a slowing of integration times for each functional form individually. **F** Violin plots showing the distribution of fit parameter values across all units from different layers of LLaMA.

units: 0.108) to 11 (median: 0.274) ($p \ll 0.001$; Mann–Whitney U test), indicating a substantial change from exponential to power-law dynamics. Since power-law functions decline much more slowly than exponential functions, this implies a substantial lengthening of integration timescales at later layers. However, we note that even at later layers the median convex combination parameter ($c$) hovered around 0.25, indicating that exponential dynamics remain present. We note that although the values of $c$ are relatively low, the overall contribution of the power-law function to the total mass of the integration window exceeds the contribution of the exponential function due to its heavier tail (by 20-fold in the final layer of GPT-2).

We also observed substantial drops in the power-law ($a$) and exponential ($b$) parameters, indicating a slower decline and thus a longer timescale. The power-law parameter showed the biggest decreases at mid-to-late layers where the contribution from the power-law fits was greatest (median value of $a$ decreased two-fold from layers 3 (1.022) to 12 (0.512); $p \ll .0001$), while the exponential parameter decreased the most in early-to-mid layers where its contribution was greatest (median value of $b$ decreased six-fold from layers 2 (4.386) to 6 (0.697); $p \ll 0.001$). Collectively, These findings demonstrate a substantial lengthening of integration timescales across LLM layers, driven by a change from exponential to power-law dynamics, as well as an increase in integration times for each functional form separately.

To assess the generality of our findings, we repeated our analyses using two different models (1): LLaMA a state-of-the-art causal LLM that is substantially larger than GPT-2 and trained on more data (Touvron et al. [2023]) (2) roBERTa a non-causal LLM trained on a masked word prediction task (Liu et al. [2019]). In LLaMA, we again observed a substantial increase in integration timescales across model layers that was well-predicted by a convex combination of an exponential and power-law (**Appendix fig. 1B**). Compared with GPT-2, the change across LLaMA's layers was primarily driven by a change from exponential to power-law dynamics with comparatively little change in the decay rates for each functional form individually (**Fig. 2F**). For roBERTa, we observed symmetric integration windows (**Appendix fig. 1D**), which when collapsed across past and future integration windows showed similar trends to those for GPT-2 (**Appendix fig. 1E**).

## 4    Assessing structure-yoked integration

The temporal extent of linguistic structures, such as sentences, is constantly varying. To account for such variation, LLMs might dynamically adapt their integration window so as to contour to structural boundaries (**Fig. 1C**). We refer to integration windows that vary with structural boundaries as "structure-yoked" and integration windows that are unaffected by structural boundaries and only depend on the distance between model and swap position ($\Delta$) as "position-yoked" (**Fig. 1D**).

To investigate whether structure-yoked integration is present in LLMs, we repeated our word-swap procedure using sequences composed of five 12-word sentences (**Fig. 3A**), in order to test whether integration windows varied with the boundary between those sentences. We again removed all punctuation, capitalization, and structure-denoting tokens (e.g., [CLS], [SEP]) to ensure there were no trivial boundary cues. We found that sentence-final words yielded slightly more tokens on average and thus restricted our analysis to sentences comprised only of single-token words to avoid this potential confound. When swapping words, we ensured that the mean embedding distance was the same for all positions in the 60-word sequence so as to guarantee that any observed differences did not reflect differences in the embedding layer. Specifically, for each swap, we sampled a desired embedding distance from a uniform distribution and then sampled a word whose distance from the original word was close to this target value when swapped in (the uniform distribution and distance tolerance were hand selected so as to provide a feasible target for the vast majority of words needing swaps; in the rare case when there was not a valid target, we sampled randomly). Sampled words were not constrained to be probable in this analysis, but results were similar using only probable swaps without the distance constraint.

We then applied our word swap procedure to the sequences, but rather than computing a single overall integration window ($\theta[\Delta]$), we computed a matrix ($\Theta[j, k]$) that encodes the change in response magnitudes as a function of both the model position ($j$) and swap ($k$) position (rather than the distance ($\Delta$) between the two) (**Fig. 3B**):

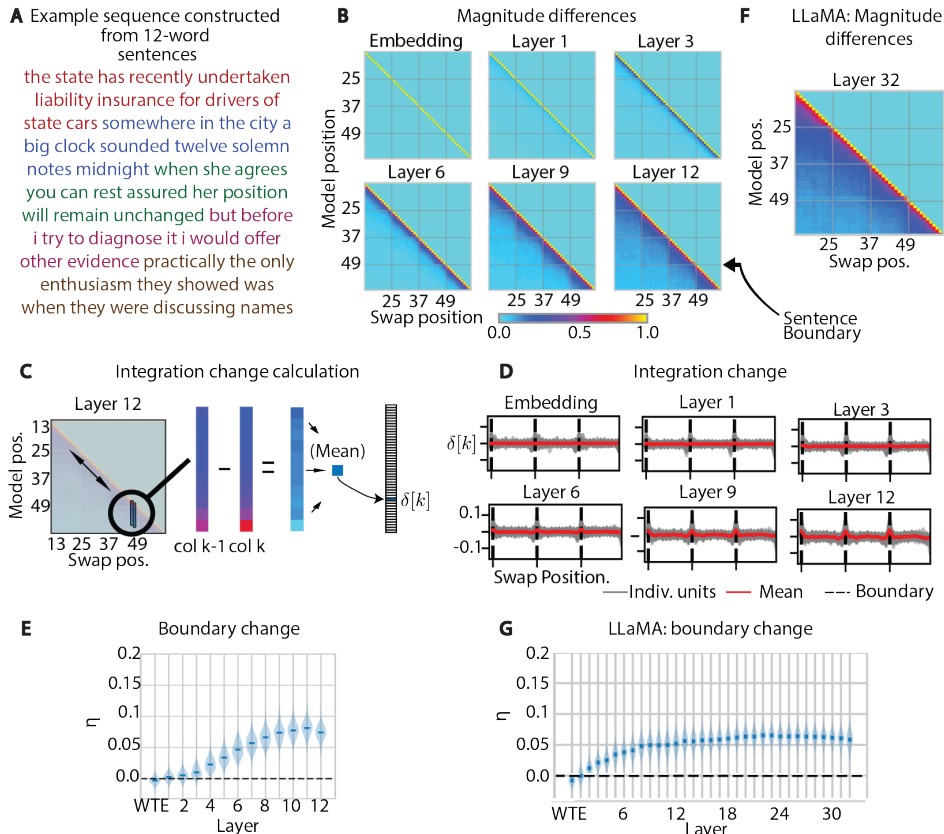

**Figure 3: Emergence of structure-yoked integration across LLM layers**. **A**, An example sequence composed of five 12-word sentences, each with unique color. No information about sentence boundaries was provided to the network. **B**, The average magnitude of response changes induced by word swaps as a function of both the model (rows) and swap (columns) position with sentence boundaries overlaid (gray lines) (for GPT-2). The emergence of structure-yoked integration is visually apparent from the triangular structure present at later network layers (see eq. 4). **C**, Schematic illustrating the calculation of the integration change metric (see eq. 5). **D**, Integration change metric for several layers of GPT-2, showing the emergence of a boundary-specific change at later network layers. Dashed lines indicate sentence boundaries. **E**, Violin plots showing the distribution of the integration change metric at sentence boundaries across units for all GPT-2 layers (see eq. 6). **F** The average magnitude of response changes for LLaMA as a function of model and swap position. **G** Integration change metric for several layers of LLaMA, showing the emergence of a boundary-specific change at later network layers.

$$\Theta[j, k] = \frac{1}{I} \sum_{i=1}^{I} |a_i[j] - a_i^{k*}[j]| \tag{4}$$

We can think of this matrix as reflecting as a time-varying integration window where $\theta_j[\Delta] = \Theta[j, j - \Delta]$ reflects the integration window at model position $j$.

As expected, when we averaged this matrix across units within a layer (**Fig. 3B**), we observed a strong effect of distance between model and swap position for all contextual layers, which manifests as a reduction in integration magnitudes for cells further from the diagonal (the diagonal corresponding to distance 0). This distance-dependent decline was slower at later layers, consistent with our previous analyses.

The second feature that is visually apparent from these matrices is a "triangular" organization that emerges at later network layers. This triangular organization is caused by a drop in the measured integration window at the boundary between sentences, which suggests the emergence of structure-yoked integration. To quantify this effect, we computed a measure of the local change in the

integration window ($N$ was set equal to one less than the structure duration, here $N = 12 - 1$; results were robust to this parameter):

$$\delta[k] = \sum_{j=k-N}^{k-1} \Theta[j,k] - \Theta[j-1, k-1] \quad (5)$$

This equation measures the average change in the integration window at a given swap position (averaged across model position) in a manner that controls for the effect of absolute distance, since the distance between the swap and model position is the same for the two quantities being subtracted. Visually, this equation corresponds to subtracting two offset, off-diagonal columns from the integration matrix (see **Fig. 3C** for schematic).

We found that the value of our integration change metric ($\delta[k]$) was close to 0, except at the boundary between sentences at later network layers (**Fig. 3D**). This boundary-specific change demonstrates that there is a drop in the integration window at the juncture between sentences in these layers, but not at other points in the sequence. This pattern suggests that the average value of our integration change metric ($\delta[k]$) at structural boundaries provides a useful measure of structure-yoked integration:

$$\eta = \sum_{k=1}^{B-1} \delta[kS + 1] \quad (6)$$

where $S$ equals the length of each structure (here $S = 12$ words) and $B$ equals the number of structures (here $B = 5$ sentences). When we plotted this boundary metric across layers (**Fig. 3E**), we observed a clear increase in structure-yoked integration from early to late network layers (the median value of $\delta[k]$ increased from 0.003 (layer 1) to 0.074 (layer 12); $p \ll 0.001$; Mann–Whitney U test). This finding suggests a transition from position-yoked to structure-yoked integration across the layers of GPT-2. A similar increase in structure-yoked integration was observed for both LLaMA (**Fig. 3G**) and roBERTa (**Appendix fig. 2B**), suggesting that the emergence of structure-yoked integration is a common feature of LLMs.

### 4.1 Generalization across different structural lengths, types, and natural contexts

We next sought to examine the generality and robustness of our effects across different types of linguistic structures and contexts.

We examined whether the degree of structure-yoked integration depends on the structure's temporal extent by varying the length of the component sentences (using our base GPT-2 model). We found that our boundary metric (6) was similar when measured for short (8 words) and long (36 words) sentences (**Fig. 4A**), demonstrating generalization across structural extent. We also tested whether a similar boundary effect could be observed for a different type of linguistic structure, by repeating our analyses using sequences composed of noun phrases (6 words) rather than sentences (specifically, noun "chunks" as computed by spaCy, which do not allow nested noun phrases within them; Honnibal and Montani [2020]). We observed a similar increase in our boundary yoking metric across network layers, demonstrating that structure yoking is not specific to sentences (**Fig. 4B**).

In our original paradigm, we concatenated randomly selected 12-word sentences and the resulting sequences therefore lack coherent linguistic content across sentence boundaries. To test whether structure-yoked integration was present at natural structural boundaries, we repeated our analyses using 84-word sequences, excerpted from paragraphs (Brown Corpus) such that each sequence had a 12-word sentence aligned at the middle of the sequence (**Fig. 4E**). We observed a clear boundary effect at the start and end of this 12-word sentence that increased across network layers (**Fig. 4D**), demonstrating that our findings generalize to natural structural boundaries. The magnitude of the boundary change was somewhat smaller than that observed for randomly selected 12-word sentences, plausibly due to greater across-sentence integration for natural sequences that have coherent across-sentence linguistic structure.

Finally, we attempted to investigate hierarchical integration of linguistic structures. Specifically, we used ChatGPT-4 to generate sequences composed of exactly 3 paragraphs, each with 3, 12-word sentences. We then calculated our boundary strength metric at paragraph and sentence boundaries

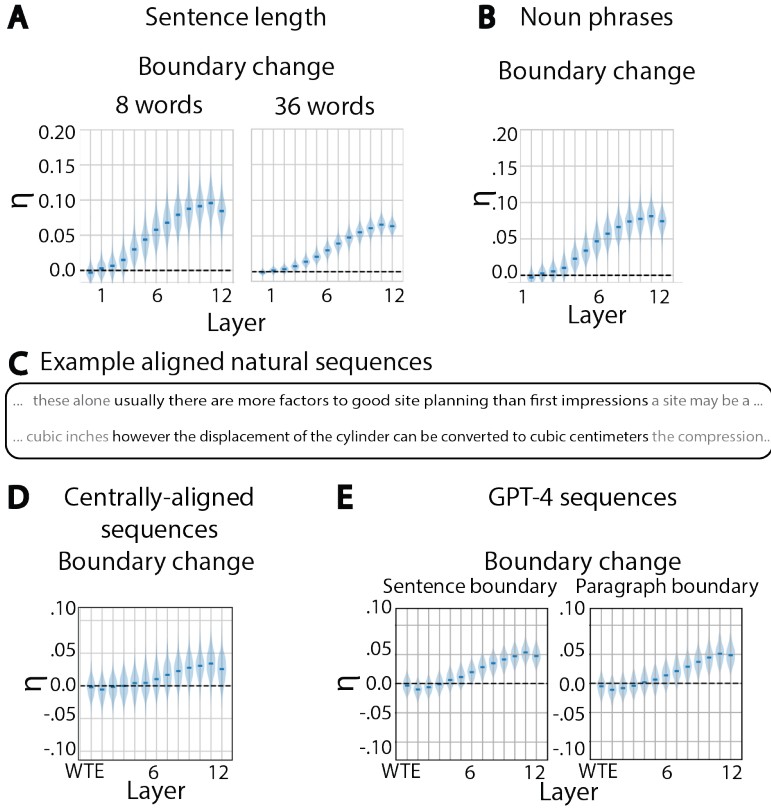

**Figure 4: Generalization across different structural lengths, types, and natural contexts**. **A**, A similar increase in structure-yoked integration was observed for sentences composed of 8 and 36 words (same format as **Fig. 3C**). Results in all panels are based on GPT-2. **B**, A similar increase in structure-yoked integration was observed for sequences composed of noun phrases instead of sentences. **C**, Examples of two sequences of words that contain a 12-word sentence aligned to the middle of the sequence (indicated by color). **D**, Boundary change metric calculated from natural sequences with centrally-aligned sentences, demonstrating the emergence of structure yoking for fully naturalistic contexts. **E**, Boundary change metric computed for hierarchically structured sequences composed of 3 paragraphs, each with 3, 12-word sentences (generated by ChatGPT-4). The boundary change metric was computed separately for paragraph boundaries and sentence boundaries (excluding paragraph boundaries). We did not observe a greater change at paragraph vs. sentence boundaries.

(excluding paragraph boundaries). We did not observe greater structure yoking at paragraph boundaries (**Fig. 4E**), in contrast with what might be expected from hierarchical integration. However, the sequences generated by ChatGPT-4 did not always have a clear topic change at paragraph boundaries, which might have limited our ability to detect hierarchical integration.

## 5   Dependence on training

We tested whether the integration windows observed in trained LLMs were learned from the structure of natural language by repeating our analyses for an untrained GPT-2 model (**Fig. 5**; using the default settings from Huggingface; Wolf et al. [2020]). We found that integration windows from the model were flat across all network layers, consistent with the idea that attention-based architectures are capable of integrating uniformly over their input (**Fig. 5A**). The integration value for $\Delta = 0$ was higher than for other distances, likely due to residual connections whose integration window is effectively a spike at 0. When we repeated our structure-yoking analyses, we found that our boundary metric was close to 0 for all network layers (**Fig. 5B**), providing further validation that our boundary metric provides a meaningful measure of structure-dependent processing in LLMs.

# 6   Conclusions, limitations, and broader significance

We have introduced a simple word-swap procedure for estimating integration windows in black-box neural language models and their dependence on linguistic structure. Using this procedure, we found that LLMs have stereotyped integration windows that substantially constrain the influence of an input word and follow a simple functional form, being well-approximated by a convex combination of an exponential and a power-law function. Across the layers of the network, we found there was a substantial increase in integration times driven partly by a transition from exponential to power-law dynamics. Finally, we found that integration windows at late network layers began to partially contour to the boundaries of linguistic structures, and we developed a metric to quantify this boundary-specific change. All of our findings were robust across different model architectures, the procedure used to swap words, and the length and type of linguistic structures manipulated, and were absent from an untrained model, demonstrating that they were learned from the structure of natural language. Thus our findings help to reveal how LLMs learn to flexibly integrate across the multiscale structure of natural language, and provide a flexible toolkit to study temporal integration in black-box language models.

Given finite time, there were limits on the number of different model architectures and linguistic structures we were able to test. Future work, for example, could examine how temporal integration windows differ between transformer models and recurrent neural networks, as well as examine a broader range of linguistic structures. While, we characterized how structure-yoked integration varies across the layers of an LLM, we did not investigate how units within a layer vary. For example, units or heads within a layer might specialize for integrating across particular types of linguistic structure, as suggested by prior work (Lakretz et al. [2019], Manning et al. [2020]).

The goal of our work was to understand existing language models, not advance the state-of-the-art. However, we believe there are many potential applications of our work. For example, one could investigate weight initialization schemes (or architectural improvements) that impose the functional forms demonstrated here at the start of training so that the network only needs to learn variations on this form (e.g., structure-yoked integration), which might improve speed or performance. Our metrics might also provide useful tools for diagnosing model limitations, such as an inability to yoke to larger-scale structures.

**Figure 5: Dependence on training**. Key findings are absent from an untrained GPT-2 model demonstrating they are learned from the structure of language. **A**, An untrained model shows flat integration windows with the exception of $\Delta = 0$. **B** An untrained model shows no evidence of structure-yoked integration.

Our methods and findings have relevance to understanding neural integration windows in biological neural systems. Currently, little is known about what functional form best describes neural integration windows in the brain and whether/how these windows vary with structural boundaries (Chien and Honey [2020], Norman-Haignere et al. [2022], in part due to methodological limitations. LLMs are state-of-art in terms of predicting human brain responses to natural language ( Schrimpf et al. [2021]), and there is considerable interest in whether the computations of these systems resemble those in the brain as well as utilizing these systems to generate new scientific insights (Anderson et al. [2021], Caucheteux et al. [2023], Tang et al. [2023]). Because our methods are model-agnostic, they are potentially applicable to measuring and modeling integration windows in biological neural systems, and our findings provide testable predictions for how neural integration windows in the brain will be structured if they are similar to those in LLMs.

## Acknowledgements

The research reported in this publication was supported by the National Institute of Environmental Health Sciences (NIEHS) and the National Institute of Deafness and Communication Disorders (NIDCD) of the National Institutes of Health (NIH) (T32ES007271 support of D.S., R00DC018051 grant to S.N.H.). The content is solely the responsibility of the authors and does not necessarily represent the official views of the NIH.

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

# Appendix

**Functional forms**

Equations for all functional forms tested is below:

- Log-Normal: $f(x|\mu, \sigma) = \exp\left(-\frac{(\ln(x)-\mu)^2}{2\sigma^2}\right)$
- Log-Cauchy: $f(x|\sigma, \mu) = \frac{\sigma}{\sigma^2 + (\ln(x)-\mu)^2}$
- Exponential: $f(x|\lambda) = \exp(-\lambda x)$
- Power Law: $f(x|\alpha) = (x+1)^{-\alpha}$
- Zipf-Alekseev: $f(x|\alpha, \beta) = (x+1)^{-\alpha - \beta \ln(x+1)}$
- Exponential + Exponential: $f(x|\lambda_1, \lambda_2, c) = c \cdot \exp(-\lambda_1 x) + (1-c) \cdot \exp(-\lambda_2 x)$
- Exponential + Power: $f(x|\lambda, \alpha, c) = c \cdot (x+1)^{-\alpha} + (1-c) \cdot e^{-\lambda x}$

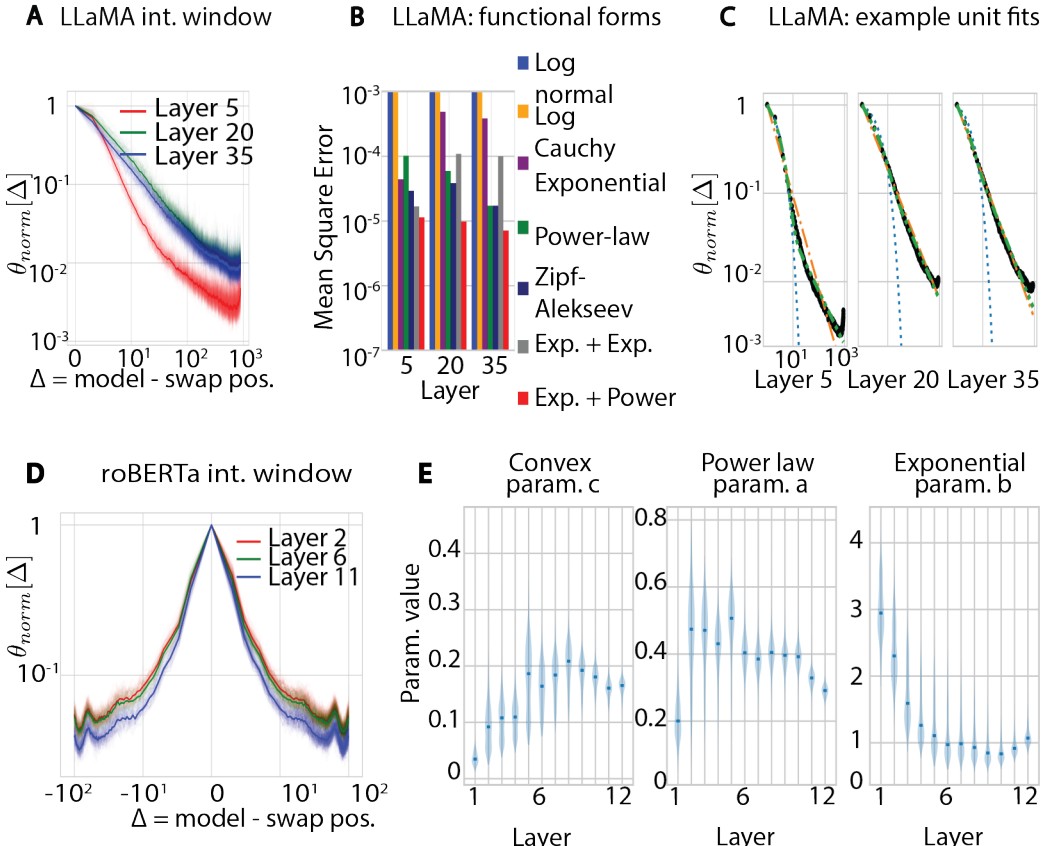

**Appendix Figure 1**. Additional plots showing integration windows for LLaMA and BERT. **A**, Integration windows for individual units from several layers of LLaMA with the median across units overlaid. The influence of a word swap decreases sharply with distance (note logarithmic scale). **B**, Quantification of the goodness-of-fit for a variety of functional forms for LLaMA. The exponential-power fits were better at predicting integration windows across all layers, as with GPT-2. **C**, Integration windows for several example units from LLaMA fit using an exponential, power-law, or convex combination of both (see eq. 3). **D**, Integration windows for individual units from several layers of roBERTa. Negative distances indicate integration across future tokens and positive distances indicate integration across past tokens. **E**, Violin plots showing the distribution of fit parameter values across all units from different layers of roBERTa. Results were very similar to those for GPT-2.

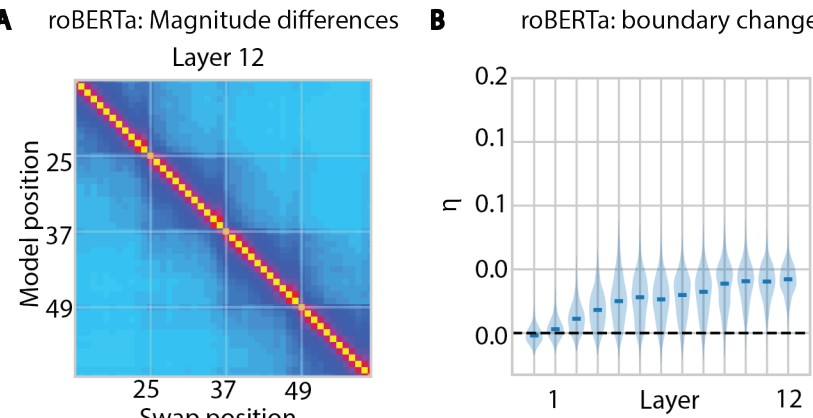

**Appendix Figure 2**. Emergence of structure-yoked integration in roBERTa. **A**, The average magnitude of response changes induced by word swaps as a function of both the model (rows) and swap (columns) position with sentence boundaries overlaid (gray lines). Results are similar to those for GPT-2, except the effects extend to both past and future tokens. **B**, Violin plots showing the distribution of the integration change metric at sentence boundaries across units for layers of roBERTa showing a substantial increase across layers, similar to GPT-2.

