# OpenReview forum: "Large language models transition from integrating across position-yoked, exponential windows to structure-yoked, power-law windows"
_NeurIPS.cc/2023/Conference — NeurIPS 2023 poster_

### Official Review · Reviewer_AsxG · 2023-07-03

**Soundness:** 4 excellent
**Presentation:** 4 excellent
**Contribution:** 4 excellent
**Rating:** 8
**Confidence:** 5

**Summary:**

The authors investigate the temporal integration window of several transformer language models (but focus on gpt2) by evaluating the effect of word swaps on the activations of individual units as a function of their distance from the swapped word. They then characterize these mean integration curves of the units in each layer as a convex combination of an exponential and power law function and highlight an evolving motif as longer context is utilized at later layers. They also explore specifically whether these windows are tied statically to only token proximity or whether they dynamically capture the boundaries of sentences (and noun phrases), and find that the integration windows of later layers are sensitive to these boundaries.

**Strengths:**

Excellent problem presentation within context of related work. The core “word swap” approach is well designed and appropriately motivated as is the analysis of “structure-yoked integration.” All experiments appear carefully constructed and controlled. The submission is clearly written and well organized. The results are important, presented clearly, and discussed appropriately.

**Weaknesses:**

The motivation of the proposed functional form is underspecified. Please include more detail as to how it was decided that the integration windows should be modeled as such. More specific question below.

**Questions:**

I would have liked to see a more principled approach starting from the integration window curves and arriving at the formulation of this convex combination of exponential and power law functions. What space of functional forms and combinations was explored? And how was this form deemed to be the best fit? Some notion of variance explained as a function of number of free parameters would be a useful datapoint to include. Just a few sentences elaborating on and clarifying this should be sufficient.

EDIT: this has now been addressed.

**Limitations:**

The limitations are appropriately addressed.

---

> ### Author Rebuttal · Authors · 2023-08-09
>
> Thank you for your supportive review.
>
> We have addressed your comment about needing to better motivate the chosen functional forms in our general response. We show that the exponential-power law function provides substantially better prediction accuracy (measured using three different metrics) than a wide range of other functional forms including exponential and power law forms in isolation.

---

> > ### Comment · Reviewer_AsxG · 2023-08-11
> > **Comments addressed**
> >
> > Thanks for including these new analyses. The quantification of model fits is convincing and the other extensions will be interesting to see. Great work!

---

### Official Review · Reviewer_NDTr · 2023-07-04

**Soundness:** 2 fair
**Presentation:** 2 fair
**Contribution:** 2 fair
**Rating:** 3
**Confidence:** 5

**Summary:**

Transformer models have the potential to acquire essentially arbitrary patterns of attention during training. But what patterns do they acquire in practice? This is the question taken up in the present paper. The data for the paper are 40 word sequences from the classic Brown corpus. The language models examined are GPT-2, LLaMA, and BERT. The paper introduces a word-swap procedure for evaluating integration. It argues that the large language models exhibit a transition from exponential to power-law dynamics across the layers of the network. It describes the power-law windows as structure-yoked (in the context of the study, this means yoked to sentence boundaries) and the exponential windows as position-yoked.

**Strengths:**

The paper raises a good question about what patterns of attention are actually acquired during the training of large language models. It undertakes to characterize the patterns in terms of functional forms. This sets up an important point of potential contact between machine learning and the study of scaling laws in physics.

**Weaknesses:**

The author(s) assert that the integration windows are surprisingly well fit by a convex combination of an exponential and a power law. They do not rigorously evaluate any alternative fits. They do not appear to be aware of the substantial research literature on power laws, and appear to have overlooked the following points.
1) Power laws can themselves be generated as mixtures of exponentials.
2) To statistically distinguish power laws from other similar-looking distributions, it would be necessary to explore a very much greater range of time scales than appear in this study.
Here are a few of the very large number of references bearing on this issue:
MEJ Newman (2005) Power laws, Pareto distributions, and Zipfs’s law. Contemporary Physics
M Mitzenmacher (2004) A brief history of generative models for power law and lognormal distributions.
RD Malmgren et al. (2008) A Poissonian explanation for heavy tails in email communications. PNAS 105(47)
S Arbesman et al (2009) Superlinear scaling for innovation in cities. Phys. Rev. E 79, 016115
Altmann et al. (2009) Bursts, Lulls, and Scaling in the Temporal Distributions of words. PLoS One 4(11

By equating structure-yoking with sentence boundaries, the author(s) disregard all other types of linguistic structures. These range from smaller components of syntactic structures (such as the structure of noun phrases) to larger structures that control discourse coherence. There seems to be no justification for thinking that all these other structures are "position yoked".

Finally the decision to report results only the 1979 Brown corpus is puzzling (the paper states that similar results were obtained using the BookCorpus, but provides no details about that). The Brown corpus contains only 1 million words, hence the vocabulary is only the more frequent words of English. It does not contain examples of various genres that figured in the training sets for the LLMs. It would be important to understand how LLMs work on the range and variety of material they were trained on.


**Questions:**

If revising this paper, it will be important to provide rigorous comparisons of different model fits, and to test the approach on a more complete range of linguistic material and a more carefully articulated set of structures.

**Limitations:**

The author(s) acknowledge that the relationship between natural language and the integration windows learned by the LLMs was not explored in depth. As mentioned under "Weaknesses", they do not appear to have a complete grasp of what is needed to justify the functional forms they selected.

---

> ### Author Rebuttal · Authors · 2023-08-09
>
> Thank you for your constructive critique.
>
> We have addressed your critique about the need for more extensive quantitative comparisons in our general response since other reviewers raised similar points. We tested a much wider range of different parametric forms motivated in part by the papers cited in your response. We find that the exponential-power form outperforms all other forms tested across multiple metrics. If there are any other specific functional forms that you would like us to test, we would be happy to do so.
>
> We are aware that power laws can be approximated as mixtures of exponentials, and will clarify this point in the manuscript. In the model comparisons described above we included a model that contains a mixture of two exponentials and thus has the same number of parameters as our exponential-power form. This alternative model performs substantially worse than our exponential-power function.
>
> Our original submission tested both sentences and noun phrases, and we find that the results are very similar across both. To address your comment and that of other reviewers we repeated our analyses using paragraphs, and again find very similar results (see general response). If there are other structures that you would consider it important to test, we would be happy to do so.
>
> To address your comment, we will include the results of BookCorpus in our revised manuscript. If there is another corpus you would like us to test, please let us know. We note that we have run many experiments investigating different architectures, swap procedures, linguistic structures, structure durations, and corpora, all of which consistently yield the same core set of results.

---

> > ### Comment · Reviewer_NDTr · 2023-08-10
> > **New calculations are improvement**
> >
> > The new calculations represent a considerable improvement in the quality of the paper.

---

### Official Review · Reviewer_hKex · 2023-07-04

**Soundness:** 4 excellent
**Presentation:** 3 good
**Contribution:** 3 good
**Rating:** 8
**Confidence:** 4

**Summary:**

This paper studies the relationship between the length of context and its influence on language model outputs across layers and units. They propose a model-agnostic procedure that swaps words at a given distance and measuring the change in the activations. Doing this for a range of distances and many different sequences allows patterns to emerge across layers. Curve-fitting shows that these patterns are a convex combination of exponential and power law functions:  largely exponential for lower layers, suggesting shorter temporal horizons, and gradually shifting to power-law for higher layers, suggesting growing temporal dependence. The parameters of the curves also suggest increasing temporal dependence within the functions. Experiments also show that these temporal horizons are bound to linguistic structure, i.e. sentence boundaries.

This paper provides empirical evidence for robustness of results by experimenting with different models and varying the details of the proposed procedures and research questions. They also provide evidence for the absence of such patterns in untrained models, which strongly suggests that the patterns discovered emerge from training on natural language.


**Strengths:**

This is an interesting study that sheds light on how language models rely on context (temporally) by analyzing changes in activations caused by swapping words at increasing distances. It provides a different and valuable perspective on potential sources of the learning signal within the data and how that may translate to certain behaviors we observe during inference. The paper makes a strong effort to reinforce results by varying the experimental setup and measuring statistical validity of the findings.

**Weaknesses:**

While the discussion is scientifically engaging, one of the frequently discussed weaknesses of analytic studies such as this one is: how can the findings of such studies provide concrete/actionable improvements and help advance the relevant fields, beyond speculation. Improving our mental models of how LMs work holds plenty of value, but if the authors have ideas or experiments on this it may be worth including them.

The experimental setup in the main experiments is a bit hard to follow. For example, what is the word replacement procedure used? How much data is used and how is it sourced? It might be worth extracting the details into a separate mini section if possible? Highlighting the findings per experiment maybe via paragraph titles may also improve clarity of the paper.

Is the data used too synthetic or too much of a toy setting? After the initial experiments, is it possible to explore a variant on more naturally occurring data such as that in language modeling datasets? Would this be feasible?


**Questions:**

Maybe I missed this in the experiments: is it hard to extend this analysis beyond sentences to contextually relevant paragraph-level analyses and demonstrate hierarchical dependence? Figure 3A includes sequences that are contextually unrelated, right?

The authors may find this older study on recurrent LMs and context interesting: https://arxiv.org/abs/1805.04623

**Limitations:**

As noted above, perhaps a discussion on the potential of this line of work to concretely influence the relevant fields could be useful. Other noted limitations seem pretty thoughtful.

---

> ### Author Rebuttal · Authors · 2023-08-09
>
> Thank you for the supportive review.
>
> We have addressed your question about the impact of our work on relevant fields in our general response because other reviewers raised similar questions.
>
> Below we have clarified the word swap procedure and the method used to select the swapped word:
>
> Word swap procedure. We first sample an N-word sequence from a corpus. We used the Brown Corpus, but the results were similar using the Book Corpus. For each word in the original sequence, we generate a new sequence with just that word swapped, yielding N sequences each with a single swapped word. We tested several methods for selecting the word to be swapped.
>
> Word selection procedure. We investigated three procedures for choosing the swapped word. (1) Part-of-speech matched. The simplest procedure randomly selected the swapped word from the set of all words with the same part-of-speech tag. (2) Probably swaps. Probable swaps were randomly sampled from a list of the 100 most probable words given the context (excluding the actual word) as computed by BERT (masking out the target word to be swapped). (3) Embedding-distance matched. The goal of this procedure was to ensure that the average embedding distance between the original and to-be-swapped word was the same for all positions, helping guarantee that any position-dependent effects (e.g., due to structure-yoked integration) could not be explained by the structure of swaps in the embedding layer. Specifically, for each swap, we sampled a desired embedding distance from a uniform distribution and then sampled a word whose distance from the original word was close to this target value when swapped in (the uniform distribution and distance tolerance were hand selected so as to provide a feasible target for the vast majority of words needing swaps; in the rare case when there was not a valid target, we sampled randomly). We found the results were very similar using all three of these procedures. We focused on the results from the simpler, part-of-speech matching procedure for our overall integration window analyses (Figure 2). We used distance matching for our structure yoking analyses since these analyses focused on position-specific effects (Figure 3), and we replicated both our overall integration window and structure yoking analyses using probable swaps (Figure 4I).
>
> We will follow your advice and clarify these methods in separate mini sections, and we will highlight our results using section and paragraph titles.
>
> We have addressed your comments about more naturalistic and paragraph-level analyses, in our general response, since a similar question was raised by other reviewers. Specifically, we repeated our analyses using paragraphs composed of three 6-word sentences, and as a consequence, the boundary between each sentence is entirely natural, unlike in our original experiments. The results from this analysis replicate our findings showing a clear change in the integration window at the boundary between sentences and also show hierarchical organization with a greater boundary change for paragraphs compared with sentences (see Figure 2). We are in the process of scaling this up to larger structures using stimuli generated by ChatGPT (e.g., 3 paragraphs each composed of 12-word sentences).
>
> The only remaining unnatural part of this procedure is the use of sequences composed of multiple fixed duration structures (e.g., 6-word sentences). To address this, we are in the process of performing the following analysis. First, we select completely natural sequences that contain a structure of a given duration (e.g., 12-word sentence) in the middle of the sequence, thus aligning the start and end of this structure across all sequences. We then repeat all of our analyses and check if there is a change in the integration window at the start and end of this single structure.
>
> Thank you for noting the Khandelwal et al. paper. We agree this is a relevant study and will reference it in our revised manuscript. Word shuffling might provide an interesting way to examine order-dependent integration in future research.

---

> > ### Comment · Reviewer_hKex · 2023-08-14
> > **Thanks for your response**
> >
> > The additional experiments and discussion are interesting and helped close some of the gaps in clarity and coverage of the initial analysis. I've raised my score to reflect this.
> >
> > By the way, in the discussion on relevance to language modeling it seems like the authors are discussing inductive bias? It wasn't super clear but might be worth a proofread or some revision for adding to the draft.

---

### Official Review · Reviewer_KfF7 · 2023-07-05

**Soundness:** 3 good
**Presentation:** 3 good
**Contribution:** 3 good
**Rating:** 5
**Confidence:** 3

**Summary:**

The authors provide an experimental understanding of LLMs that how LLMs have inherent Integration windows to take account of the global meaning of the given sentence, by developing a novel method called “word-swap procedure,” which is model agnostic. The authors investigate the behavior of the Integration window with various control: changing the inner layer, changing the distance from the current position of sentence tokens, or varying sentence structures. The finding is that some trained LLM’s Integration window fit well with a convex combination of an exponential and power law function, resulting in the exponential law window at early layers across position-yoked, and the power law window at later layers followed by structure-yoked.

**Strengths:**

- Suggest a metric for quantifying the variance of integration windows
- propose model agnostic analysis method to understand the integration window inherent LLMs
- provide an explicit experimental understanding of how LLM’s integration window behaves across layers, and with varying sentence structure.

**Weaknesses:**

- It seems that the kinds of tested LLMs are quite restricted (only three), which bound the extensiveness of the analyses, let alone that GPT-2 is actually not that large with respect to current LLMs. In addition, for the generalization of the transformer architecture, it seems necessary to add the result of the encoder-decoder transformer.
- The number of tokens for each sequence seems to be fixed at most 40, but the methods author proposes can be applied to the longer sequence length. It would be helpful to understand the property of the integration window if the author provides enriched experiments with longer sequence lengths.
- Though the authors exhibit that the integration window of LLM shows different behavior across the layers, that the lower layer follows the exponential law, and that the higher layer follows the power law, the contribution itself seems to be not enough.

**Questions:**

- Can you specify what kind of linguistic structure is used for the experiments of structure-yoked integration in Figure 3?

**Limitations:**

- Integration window may not appear only for simple text generation, but could be observed in other tasks of NLU, or summarization, or text retrieval (though summarization and text retrieval are belong to generation, these tasks would need more detailed behaviour of Integration window). Could you provide experimental investigations other than simple text generation?

---

> ### Author Rebuttal · Authors · 2023-08-09
>
> To address your comments, we will test additional, larger models including LongT5. LongT5 is an encoder-decoder architecture that can accept very long sequences and has been trained on multiple tasks, addressing each of the issues that you raised in your review. The T5 model has also shown strong neural predictivity in the brain (Schrimpf et al. (2021) The neural architecture of language: Integrative modeling converges on predictive processing) (LongT5 has not been tested to our knowledge). We plan extend our analyses to much longer sequences (e.g., 1000 tokens).
>
> We note that our contribution extends beyond simply showing a transition from exponential to power law dynamics. In particular, a key result of our study is that late layers adapt their integration window to structural boundaries in language, while earlier layers do not (or do so weakly). This suggests there is a transition from position-yoked to structure-yoked integration in LLMs, a finding that we replicate across all models tested. Our study also introduces new methods for measuring both the overall integration window and structure yoking.
>
> We used 12-word sentences for our analyses in Figure 3. Figure 4K shows the results for 8- and 36-word sentences and Figure 4L shows the results for 6-word noun phrases. In our general response, we describe the results of a new analysis using paragraphs composed of three 6-word sentences. All of these analyses qualitatively show the same effect and our analyses with paragraphs further reveal hierarchical structure yoking. We will clearly label the structure tested in the caption of all figures.

---

> > ### Comment · Reviewer_KfF7 · 2023-08-21
> >
> > Thank you for the author's detailed response.
> > After reading the author's reply and the comments from other reviewers, I will maintain my score.

---

### Official Review · Reviewer_3ifa · 2023-07-06

**Soundness:** 4 excellent
**Presentation:** 4 excellent
**Contribution:** 3 good
**Rating:** 8
**Confidence:** 5

**Summary:**

This paper describes a novel method for measuring and characterizing integration behavior in large language models. This method is used first to look just at temporal integration windows—i.e. how do inputs at different lags affect model activations at a specific time—in GPT-2, revealing a transition from exponential-like to power-law-like behavior across model layers. Second, the authors investigate whether integration is “position-yoked” or “structure-yoked” by performing a similar analysis on strings comprising multiple concatenated sentences. These results show that the degree of structure-yoking, like integration window size, increases consistently across model layers. Finally, these results are replicated on other networks (LLaMA and roBERTa) and with some variations in language input.


**Strengths:**

This is overall an interesting and well-executed paper, with many strengths:
* The procedures are well-motivated theoretically, clearly explained, and obviously well-suited to measuring the effects of interest.
* In particular, the structure-yoking experiment is cleverly constructed and tests a very interesting property of these models.
* Evaluation is fairly exhaustive, with tests of many networks and different variations on the procedure to ensure robustness.
* The paper makes contact with much of the relevant literature, linking the work both to machine learning and neuroscience research.

**Weaknesses:**

* My only actual issue with the paper is that it does not do a terribly good job at answering the question, “so what?” This seems like timely and interesting work, so the authors should be able to say a little about why it’s useful to deeply understand and characterize integration behavior in these networks.
* Differences between exponential and power law decay are quite difficult to see with linear scales; the authors should try showing the relevant data (especially Figure 2C) using log-log plots.

**Questions:**

* The analyses consider structure only at the scale of noun phrases (6 words) and sentences (8 to 36 words). What about higher-order structure? Do the same layers also “yoke” to structure at supra-sentential scales, like paragraphs or (maybe possible only with LLaMA) entire narratives?

**Limitations:**

The discussion of limitations is clear and complete.

---

> ### Author Rebuttal · Authors · 2023-08-09
>
> Thank you for the supportive review.
>
> We chose to address your questions about the impact of this work and the generalization of our paradigm to supra-sentential scales in our general response since similar questions were raised by other reviewers.
>
> We now plot integration windows on a log-log scale (see Figure 1 of included PDF).

---

> > ### Comment · Reviewer_3ifa · 2023-08-14
> >
> > Thanks to the authors for the detailed responses & new analyses. In particular, I think the authors did a good job addressing the question of significance. I already thought this paper was good, and I still think it's good.

---

### Author Rebuttal · Authors · 2023-08-09

We were pleased the reviewers overall felt that our work addressed a timely question, that the methods were well-motivated and described, and that the results were interesting and robust across multiple experiments. We thank the reviewers for their constructive critiques. Below, we address comments shared across multiple reviewers.

Functional form of integration windows

Multiple reviewers requested additional motivation/quantification of the functional form used to model integration windows. To address this issue, we performed several analyses, making the following changes:

1.	Tested additional functional forms motivated by prior literature (including papers noted by reviewer NDTr): (1) exponential (2) power (3) exponential-power (4) exponential-exponential (5) Zipf-Alekseev (6) log-normal, (7) log-Cauchy.
2.	Quantified goodness-of-fit with multiple metrics: (1) cross-validated mean-squared error (MSE), (2) cross-validated Kolmogorov-Smirnov (KS) test statistic (3) Bayesian Information Criterion (BIC).
3.	Extended the sequence length from 40 words to 150 words (we plan to extend it further, e.g. 1000 words).
4.	We plot integration windows on a log-log scale to better visualize the tail.

We found that the exponential-power form provides the best fit across all GPT-2 layers using all 3 metrics (see Figure 1) (we will repeat this analysis for the other tested models).

Interestingly, when plotted on a log-log scale, integration windows appear to exhibit piecewise-linear decay with a single knot (Figure 1). We are testing whether a piece-wise linear form, corresponding to a transition between two power laws, shows even better fits. The results of this analysis will not change our finding that LLM integration windows can be approximated using a simple (3-parameter) functional form whose timescale substantially expands across layers. It also has no impact on our structure yoking findings or methodological contributions (word swap procedure, structure yoking paradigm, and boundary metric). We will revise the manuscript to incorporate all of these results.

Larger-scale hierarchical structure

Several reviewers (3ifa, hKex, NDTr) asked if our structure-yoking results would occur at supra-sentential scales and/or exhibit hierarchical organization, as well as whether structure yoking would be evident at natural boundaries between sentences (as opposed to between randomly selected sentences). To investigate these questions, we repeated our analyses using paragraphs composed of three 6-word sentences (the longest sentence length for which we could find enough paragraphs) (Fig 2). We observe structure yoking at the boundary between sentences, demonstrating that our effects generalize to natural boundaries. We also observe even stronger yoking to the boundary between paragraphs, suggesting hierarchical integration. We are working on scaling up these analyses by using ChatGPT (GPT-4) to craft larger-scale structures with stereotyped durations (e.g., 3 paragraphs each with 12-word sentences), as well as repeating these analyses with larger LLMs (e.g., LLaMA, LongT5). We will revise the manuscript to include these new results.

Significance to the NeurIPS community

Multiple reviewers asked us to better articulate the significance of our work for the NeurIPS community:

Relevance to neuroscience. Understanding how the human brain integrates linguistic information is an important question of active interest to the NeurIPS community (e.g., Jain et al. (2020) Interpretable multi-timescale models for predicting fMRI responses to continuous natural speech). The brain must have mechanisms for integrating flexibly across multiple timescales and linguistic structures. Yet, it is largely unknown what functional form best describes human cortical integration windows and whether/how these windows vary with structural boundaries, in large part due to methodological limitations. LLMs are state-of-art in terms of predicting human brain responses to natural language, and there is considerable interest in whether the computations of these systems resemble those in the brain and utilizing these systems to generate new scientific insights (Caucheteux et al (2023) Evidence of a predictive coding hierarchy in the human brain listening to speech, Tang et al. (2023) Semantic reconstruction of continuous language from non-invasive brain recordings). Because our methods are model-agnostic, they are directly applicable to measuring and modeling integration windows in biological neural systems. Our findings provide clear, testable predictions for how neural integration windows in the brain will be structured if LLMs integrate information in a brain-like manner.

Relevance to language modeling. The goal of our work was to understand existing language models, not advance the state-of-the-art. We agree that our paper does not provide specific guidance on how to improve language models, which we will note as a limitation. The impact of empirical insights on applied research is often difficult to predict, and we believe there are many potential applications of our work. For example,  LLM integration windows show a stereotyped functional form that differs substantially from untrained networks and is robust across different architectures. Thus, a potentially interesting research direction would be to investigate weight initialization schemes (or architectural improvements) that impose this functional form so that the network only needs to learn variations on this form (e.g., structure-yoked integration), which might improve speed or performance. Our metrics might provide useful tools for diagnosing model limitations, such as an inability to yoke to larger-scale structures. We will briefly note these potential research directions/applications in the revised manuscript, flagging them as speculative.

---

### Decision · Program_Chairs · 2023-09-21

**Decision:**

Accept (poster)

**Comment:**

The work presents a method for measuring and characterizing integration behavior in LLMs, focusing on temporal integration windows and structure-yoking.

The reviewers collectively acknowledge several strengths of the paper:
- The paper is well-motivated theoretically, and the procedures are clearly explained, making it suitable for measuring the effects of interest in LLMs.
- The experiments, particularly the structure-yoking analysis, are cleverly designed and explore an intriguing property of LLMs, providing valuable insights.
- The paper offers an exhaustive evaluation, testing multiple networks and variations to ensure robustness in its findings.

However, reviewers express some concerns and suggest areas for improvement:
- The paper tests a relatively small set of LLMs, which limits the breadth of the analysis. Reviewers suggest including more diverse LLM architectures, such as encoder-decoder transformers.
- The fixed sequence length of 40 tokens may not capture the full picture of integration behavior. Expanding experiments with longer sequence lengths could provide valuable insights.
- Some reviewers seek a clearer explanation of the practical significance of the research beyond theoretical insights. They suggest exploring potential applications and improvements for LLMs.
- There are concerns about the synthetic or toy-like nature of the data used in experiments. Reviewers propose experimenting with more naturally occurring data, like language modeling datasets, for broader applicability.
- The choice of the exponential-power functional form for modeling integration windows requires more detailed motivation and quantification of its appropriateness.

The authors' response effectively addresses these concerns by conducting additional analyses with various functional forms, quantifying goodness-of-fit, extending sequence lengths, and exploring hierarchical structure yoking. They also articulate the significance of their work for the NeurIPS community, emphasizing its relevance to both neuroscience and language modeling. These rebuttal details should be included in the final paper draft.